# Blockchain-Empowered Digital Twins Collaboration: Smart Transportation Use Case

**Radhya Sahal** [1,2,*] , **Saeed H. Alsamhi** [3,4], **Kenneth N. Brown** [1], **Donna O'Shea** [5], **Conor McCarthy** [6] **and Mohsen Guizani** [7]

1  SMART 4.0 Fellow, School of Computer Science and Information Technology, University College Cork, T12 E8YV Cork, Ireland; k.brown@cs.ucc.ie
2  Faculty of Computer Science and Engineering, Hodeidah University, Al Hodeidah 3114, Yemen
3  SMART 4.0 Fellow, Software Research Institute, Athlone Institute of Technology, N37 W089 Athlone, Ireland; salsamhi@ait.ie
4  Faculty of Engineering, IBB University, Ibb 70270, Yemen
5  Department of Computer Science, Cork Institute of Technology, T12 P928 Cork, Ireland; donna.oshea@cit.ie
6  School of Engineering, University of Limerick, V94 T9PX Limerick, Ireland; conor.mccarthy@ul.ie
7  College of Engineering, Qatar University, Doha 2713, Qatar; mguizani@ieee.org
*  Correspondence: rsahal@ucc.ie

**Abstract:** Digital twins (DTs) is a promising technology in the revolution of the industry and essential for Industry 4.0. DTs play a vital role in improving distributed manufacturing, providing up-to-date operational data representation of physical assets, supporting decision-making, and avoiding the potential risks in distributed manufacturing systems. Furthermore, DTs need to collaborate within distributed manufacturing systems to predict the risks and reach consensus-based decision-making. However, DTs collaboration suffers from single failure due to attack and connection in a centralized manner, data interoperability, authentication, and scalability. To overcome the above challenges, we have discussed the major high-level requirements for the DTs collaboration. Then, we have proposed a conceptual framework to fulfill the DTs collaboration requirements by using the combination of blockchain, predictive analysis techniques, and DTs technologies. The proposed framework aims to empower more intelligence DTs based on blockchain technology. In particular, we propose a concrete ledger-based collaborative DTs framework that focuses on real-time operational data analytics and distributed consensus algorithms. Furthermore, we describe how the conceptual framework can be applied using smart transportation system use cases, i.e., smart logistics and railway predictive maintenance. Finally, we highlighted the future direction to guide interested researchers in this interesting area.

**Keywords:** blockchain; digital twins; Industry 4.0; smart manufacturing; data analysis; transportation; logistics; railway

## 1. Introduction

Industry 4.0 revolution is considered to be a new paradigm of digital, autonomous, and decentralized control with the Industrial Internet of Things (IIoT), Machine Learning (ML), big data, and edge computing [1,2]. For smart manufacturing, distributed manufacturing is a form of decentralized manufacturing practiced by enterprises using digitalization [3,4]. It uses an effective collaboration form in terms of information sharing, analytics, and collaborative decision-making in real-time. A Digital Twin technology (DT) is one of the core elements of manufacturing digitalization, representing a real-world system such as production systems in a virtual space [5]. Multiple DTs are used to represent a distributed production system in hierarchical levels [6]: (i) DTs in a flat network represent individual things at the machine level. They exchange information with each other on things and learn about their operation and behavior to build a common understanding of

the machine condition, (ii) DTs for things in a tree or a chain represent the sub-system level and the system level where each DT is passing on information to the next level.

Multiple DTs are deployed to represent the up-to-date industrial data of the physical assets in operation, including the asset status and the relevant historical data. The deployed DTs can intelligently collaborate by utilizing the intelligence of DT-driven operational data to predict the potential risks within the distributed manufacturing systems. In particular, the DTs are collaborating by applying predictive data analytics to analyze DT-based historical operating data, learn about their things using shared knowledge and real-time data, and then predict the potential risks in real-time. A better understanding of the predicted potential risks can facilitate consensus decision-making among participants on the management floor within the distributed manufacturing systems. However, the DT paradigm is still at an early stage, and many challenges still exist to adopt DTs collaboration in the distributed manufacturing environment, including:

- Interoperability: The models and strategies of the sharing policies (i.e., internal and external data) need to define the DTs data schema and the collaboration requirements.
- Authentication: In some scenarios in distributed manufacturing systems, the deployed DTs are owned by independent entities that want to collaborate. Therefore, securing a digital distributed manufacturing system needs efficient technology to acquire secure real-time data exchange and analysis across multiple participants.
- Distributed machine learning: A large-scale input data size from multiple participants needs to be analyzed to obtain accurate predictions about the potential risks within the distributed manufacturing system.
- Distributed decision-making: Centralizing suffers from single failure data, while decentralization suffers from lacking global data, so the decision-making consensus is required.
- Scalability and robustness: a system needs to accommodate a large number of DTs which represent multiple participants, e.g., objects, devices, machines, nodes, people, workstations, etc., within manufacturing systems. The distributed manufacturing system also needs to deal with multiple deployed DTs and simultaneously maintain the robustness at a required level, especially with hacked nodes and malfunctioning.

Most works have proposed adopting blockchain with DTs to guarantee transparency, decentralized data storage, data sharing, peer-to-peer communication, secure and trusted traceability, and scalability [7]. Using blockchain, multiple DTs can collaborate in a hierarchical and granular manner, using shared knowledge to manage and trace the product assembly data [8]. A smart contract is used to execute some actions automatically to increase data sharing efficiency, and higher security [9], and provide trusted data provenance, audit, traceability, and tracking transactions initiated by participants involved in the creation of DTs [10]. However, the existing research lacks solutions for collaborative DTs based on operational data analytics because its focus is mainly related to blockchain adoption for DTs. There are still many challenges requiring further investigation to identify, diagnose, and remove the potential risks in distributed manufacturing systems using the intelligence of the DT-driven operational data.

### 1.1. Contribution

The combination of blockchain and DT technologies represents the key technologies that allow continuous data acquisition in Industry 4.0. Furthermore, the combination of both technologies has significant advantages to address the challenges mentioned above such as traceability, security, the guarantee of ownership rights, decentralization, etc. [7,8,10–12]. However, the combination of blockchain and DT is rather still under exploration. Many research works have proposed simple blockchain model adoption of DT with a focus on centralized production systems. In contrast, these works have fallen short of providing DT-based solutions for distributed manufacturing. Moreover, the blockchain is not designed for DTs collaboration scenarios of risk prediction. On the other hand, the distributed consensus decision-making has been adopted in blockchain technology [13,14].

Various consensus implementations were proposed to make replicas reach an agreement on transactions updating using a distributed ledger. However, the development of many ledger-based DTs with the distributed consensus decision-making for risk prediction is still an unsolved problem. Therefore, more smart and collaborative solutions for DTs based on Distributed Ledger Technology (DLT) and distributed consensus decision-making for risk prediction are required to add progressive value to distributed manufacturing.

Consequently, the main purpose of this work is to develop a conceptual framework for data-driven ledger-based collaborative DT. In particular, the proposed framework targets smart distributed manufacturing to predict the potential risks using the intelligence of sharing operational data. Figure 1 depicts the high level of the blockchain-based collaborative DTs using predictive data analysis (i.e., analysis DT-based operational data).

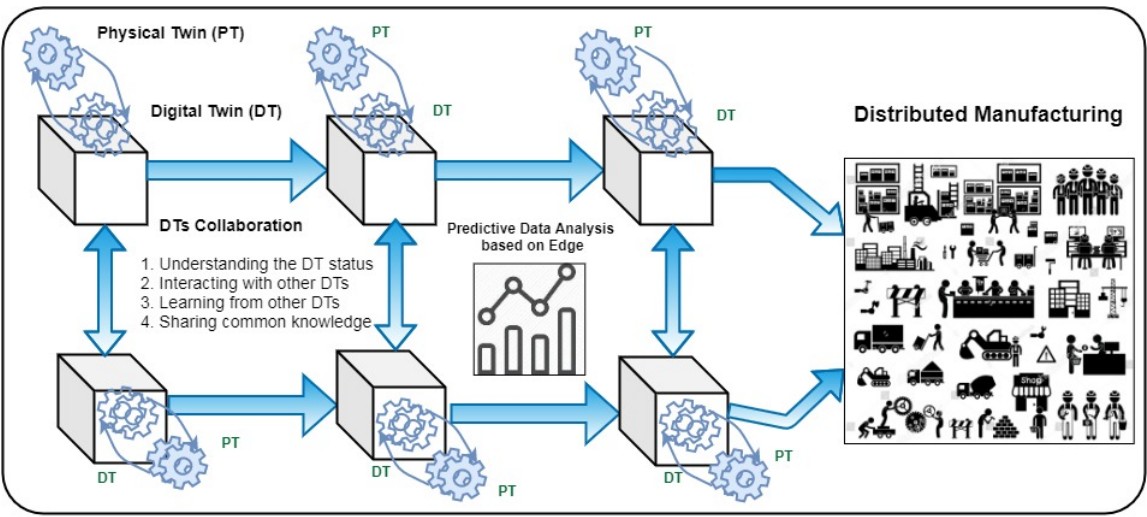

**Figure 1.** The high-level of the blockchain-based collaborative digital twins (DTs) using predictive data analysis (i.e., analysis DT-based operational data).

Our main contributions in this conceptual framework paper are summarized as follows:

- We explore how blockchain employing in DTs collaboration with highlighting the benefits of the combination.
- We propose the conceptual framework of the data driving-based DTs collaboration with the help of blockchain technology. The proposed framework consists of two components:
  1. The data-driven ledger-based predictive model is used to predict the potential risks using DT-driven operational data. The DLT performs intelligent and secure interoperability, including real-time operational data exchange, querying the real-time operational database, and dynamic interactions among the deployed DTs. At the same time, the distributed predictive model plays a vital role in developing and evaluating DT deployment locally using the DT-driven operational data.
  2. A distributed consensus algorithm to improve the decentralized DTs collaboration. The distributed decision-making algorithm develops based on the essence of the consensus mechanism and the dynamic prediction, which uses real-time DT-driven operational data. The developed distributed consensus algorithm can make most nodes agree on the potential risks and notify the decision-makers within the distributed manufacturing systems.
- We describe how the conceptual framework can be applied in smart transportation systems, i.e., smart logistics and railway predictive maintenance.

### 1.2. Paper Organization

The remainder of this paper is organized as shown in Figure 2. The comparison with other existing solutions of blockchain empowering digital twins collaboration is introduced in Section 2. The proposed conceptual framework for collaborative DTs is introduced in Section 3. The use case of smart transportation is described in Section 4. The discussion, validation, and future direction are presented in Section 5. Finally, conclusions are presented in Section 6.

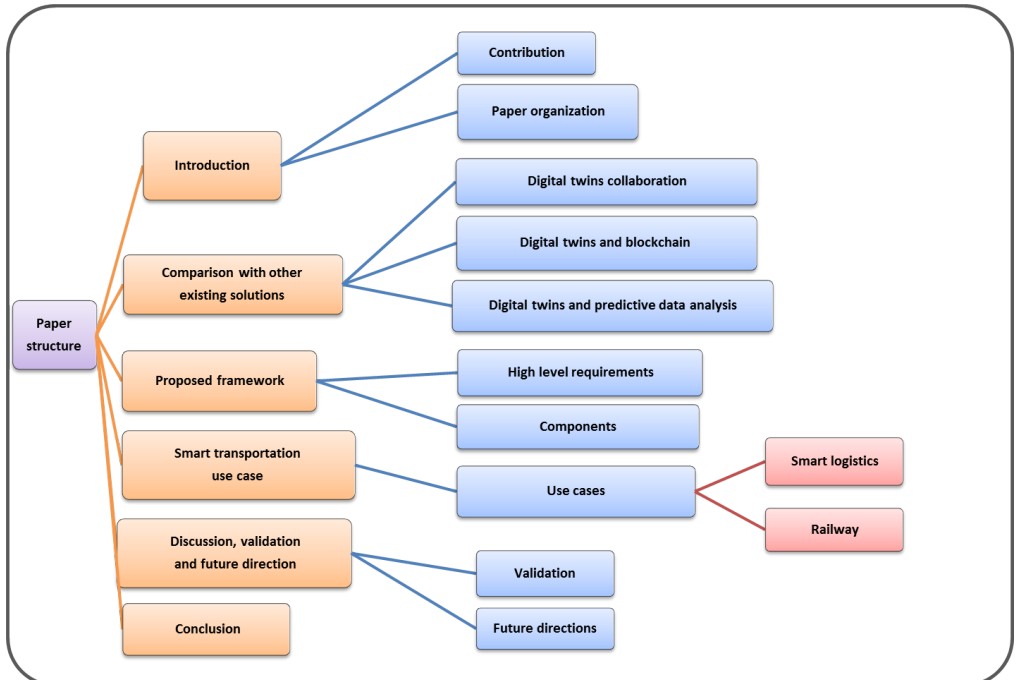

**Figure 2.** Paper structure.

## 2. Comparison with Other Existing Solutions of Blockchain Empowering Digital Twins Collaboration

In this section, we compare the contributions of our work with state-of-the-art solutions, particularly solutions based on the use of blockchain and digital twins.

### 2.1. Digital Twins Collaboration

DTs are merging the virtual worlds and real worlds. It is used to describe the detailed presentation of machines, devices, robots in the warehouse, production, and process. The DTs' advantages in Industry 4.0 include improving data security and quality, reducing cost, and faster decision-making. The authors of [15] described DTs as one virtual replica of a machine, robots, and devices containing data, function, and communication interfaces. The main parts of DTs are the physical entity, virtual entity, and information that connect virtual and physical entities [16,17] as shown in Figure 3 [18].

Collaboration means sharing and exchange information among entities and share tasks to act accordingly. The authors of [19,20] discussed the importance of Artificial Intelligence (AI) and Machine Learning (ML) for robots. The collaboration is based on improving the quality of services, connectivity, and reliability. Furthermore, the collaboration of drones and the Internet of things to enhance smartness of smart cities applications [21] and public safety [22], and for better Quality of Service (QoS) [23]. The collaboration among multi-user and identifying the activities is described in [24]. Collaboration of DTs and humans is described with details in [25]. However, the authors highlighted the challenges of collaboration in industry platform [26–28].

Collaboration is essential for a group of users to perform complex activities effectively and efficiently, while a single can not do [24]. In [29], the authors introduced blockchain

technology for heterogeneous multi-robot collaboration to combat COVID-19 in decentralized peer-to-peer networks without human intervention. Furthermore, the authors of [30] applied blockchain for decentralized multi-drone collaboration to combat COVID -19 in delivering goods and monitoring people in the quarantine area. Furthermore, the authors [19] introduced a machine learning technique for multi-robot collaboration based on keeping connectivity, maintaining the quality of services, and improving mobility during task performances. In [22] addressed drones and IoT devices collaboration to improving greener and smarter cities, while the drones and IoT collaboration resulting green IoT [31,32]. However, no one of the above studies addresses DTs collaboration and applications. Therefore, we discuss DTs collaboration for improving transportation and logistics applications.

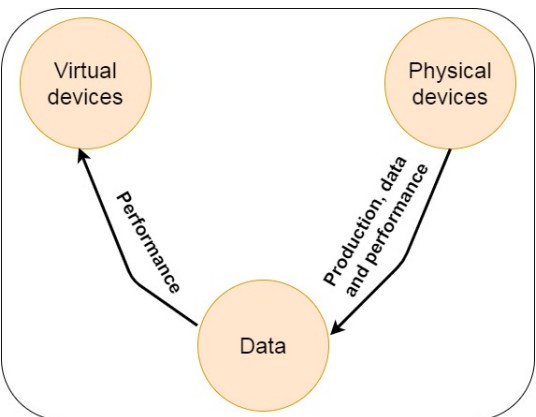

**Figure 3.** The main parts of DTs [18].

Smart industries depend on gathered data from smart IoT devices or their DTs of the production lines. Collected data can be erratic sensors, RFID, actuators, or their DTs injecting and producing incorrect data for analyzing and taking affecting decision-making. Based on this, the author [18] has proposed a unified interaction mechanism for collaborative DTs to provide an auto-detection using the intelligence of operational data in the cyber-physical production system. The proposed mechanism can detect whether the DT has erratic behavior by interacting with other collaborative DTs within the edge level. The authors [33] introduced architecture for smart factories, relying on service-based DTs. The architecture was presented how automatically DTs combine the corresponding physical processes and sharing analogies with web services. The authors of [34] introduced DT and big data in the smart industry, focusing on applications, manufacturing, production, maintenance prediction, etc. Furthermore, the authors of [35] discussed the enabling technologies for DT in smart industries. He et al. [36] presented DT-driven sustainable smart industries. Moreover, the authors of [37] introduced DTs with Industry 4.0 and data analytics.

### 2.2. Digital Twins and Blockchain

DT technology has been aligned with blockchain technology in different industrial sectors [7]. ManuChain is an iterative bi-level model proposed based on the incorporation of blockchain into DT on decentralized manufacturing [11]. ManuChain model has defined the lower-level for fine-grained self-organization intelligence and upper-level to iterate the coarse-grained holistic optimization intelligence by utilizing decentralized features of blockchain. Makerchain is another blockchain-driven model based on DTs, which was proposed to handle the cyber-credit of social manufacturing among various makers [38]. The Makerchain model has used DTs to synchronize the updating of data tags and ensure the personalized demands to make manufacturing service transactions among makers more trustworthy. In [39], the authors have proposed a manufacturing blockchain of things (MBCoT) architecture for secure, traceable, and decentralized manufacturing configuration. The authors have defined the data-and knowledge-driven DT manufacturing

cell as a reference model for decentralized manufacturing. They have also introduced the consensus-oriented transaction logic of MBCoT for the fault-tolerant protocol to support the autonomous manufacturing process. In [40], the authors have developed ExplorerChain, including online ML, metadata of transaction, and the Proof-of-Information-Timed algorithm to serve as a reference for researchers who would like to implement and deploy blockchain technology in the healthcare/genomic domain. In [12], the authors have used DLT to develop a protocol to guarantee the transfer of values between DTs in economic systems. Furthermore, a framework for secure DT data sharing based on DLT is proposed to track events and provenance information along with an asset's lifecycle and increase transparency for all participants [41].

All of the above literature discussed collaboration in smart industries. Still, none of the studies focused on discussing the collaboration of DTs using blockchain technologies to improve management in the distributed smart manufacturing.

### 2.3. Digital Twins and Predictive Data Analysis

A predictive DTs methodology has been proposed to enable data-driven physics-based DTs using a library of component-based reduced-order models and interpretable machine learning [42]. The predictive DTs methodology has been performed using a case study of fixed-wing UAVs. The UAV uses structural sensors to detect damage or degradation on one of its wings to inform the UAV's decision-making about performing an aggressive maneuver or a more conservative one to avoid structural failure. In [43], a smart city DT architecture is proposed where knowledge representation, reasoning, and ML formalisms are used to provide complementary and supportive roles in the collection and processing of data, identification of events, and automated decision-making.

Maintenance is one of the predictive data analysis-based applications in Industry 4.0, which is referred to as predictive maintenance 4.0 or PM 4.0. It is one of the recent researched industrial applications as its impact on the manufacturing sectors. The authors in [44] have reviewed the use of DTs technologies to apply maintenance strategies to provide a deeper insight into the synergies between both DTs and maintenance in the industrial sectors. Furthermore, the authors in [1] proposed a set of requirements to enable predictive maintenance with big data for Industry 4.0 applications. The authors have studied the railway industry concerning big data streaming processing platforms, distributed message queue management systems, big data storage platforms, and streaming SQL engines. Moreover, the authors in [45] have proposed a ledger-based DT reference model for predictive maintenance defined in three layers: edge, fog, and cloud. Regarding data analysis of DTs, Song et al., introduced a predictive maintenance model that comprised DTs plurality [46]. DTs corresponded to the plurality of remotely located physical machines. Each DT combined product nameplate data that correspond to a unique physical device with more simulation models. The database contained the run time and logged data gathered from sensors with the associated physical machine. Moreover, modular-based corrective maintenance was used in DTs, which was proposed for automating decision-making in complex systems [47]. The proposed model was corrective maintenance and was relied on DTs development. Aivaliotis et al., proposed the physics approach to predict the maintenance by using DTs [48]. The proposed approach was discussed to reduce modeling efforts and provide the modeling framework of different resources for enabling DTs. The machine modeling was included the machine dynamic behavior using the grey box, black box, and white box. The virtual sensor modeling referred to gather data during the simulation. Furthermore, modeling parameters were included parameters updated with frequency to guarantee machine DT.

Table 1 describes a comparison of existing work and the present work concerning the applications, including blockchain, DTs, collaboration, data analysis, Industry 4.0, and transportation. Because only a few publications exist in blockchain-based DT, DT collaboration has not yet been a focus in the literature to date. Moreover, data interoperability-by-design concepts have not been considered yet.

**Table 1.** Comparison of existing work and the present work (Industry 4.0 (I4.0), Transportation(Transp)).

| Ref | Highlighted | Blockchain | DT | Collaboration | Data Analysis | I4.0 | Transp |
|---|---|---|---|---|---|---|---|
| [11] (2020) | ManuChain model is proposed based on the incorporation of blockchain into DT on a decentralized manufacturing | ✓ | ✓ | X | X | ✓ | X |
| [38] (2019) | Makerchain model based on DTs is proposed to handle the cyber-credit of social manufacturing among various makers | ✓ | ✓ | X | X | ✓ | X |
| [40] (2020) | ExplorerChain is a reference for researchers to implement and deploy blockchain technology in the healthcare/genomic domain using online machine learning | ✓ | X | X | ✓ | ✓ | X |
| [39] (2020) | MBCoT architecture for the configuration of a secure, traceable, and decentralized manufacturing | ✓ | X | X | X | ✓ | X |
| [12] (2019) | A protocol based on DLT to guarantee the transfer of values between DTs in economic systems | ✓ | ✓ | X | X | ✓ | X |
| [29] (2020) | The homogeneous and heterogeneous of multi-robot collaboration to perform their complex task in decentralized fashion with blockchain technology | ✓ | X | ✓ | X | X | X |
| [45] (2019) | Aledger-based DT reference model for predictive maintenance | ✓ | ✓ | X | ✓ | ✓ | X |
| [41] (2019) | A framework for secure DT data sharing based on DLT to track events and provenance information | ✓ | ✓ | X | X | ✓ | X |
| [42] (2020 ) | A predictive DTs methodology based on ML using UAV case study | X | ✓ | X | ✓ | ✓ | X |
| [43] (2020) | A smart city DT architecture using reasoning and ML to automated decision-making | X | ✓ | X | ✓ | ✓ | X |

**Table 1.** *Cont.*

| Ref | Highlighted | Blockchain | DT | Collaboration | Data Analysis | I4.0 | Transp |
|---|---|---|---|---|---|---|---|
| [19] (2019) | Machine learning technique for multi-robot collaboration based on keeping connectivity, maintaining the quality of services, and improving mobility during tasks performances | X | X | ✓ | ✓ | X | X |
| [22] (2019) | Drones and IoT devices collaborate to improving greener and smarter cities | X | X | ✓ | X | X | ✓ |
| [49] (2018) | DT monitoring that for monitoring and development of wind farms. | X | ✓ | X | ✓ | X | X |
| [50] (2017) | An approach for identifying the network physical vulnerabilities in industry 4.0 systems. | X | X | | X | ✓ | X |
| [51] (2014) | AI-based supervisory control and data acquisition method for prediction and fault diagnosis of wind turbines | X | X | X | ✓ | X | X |
| [33] (2019) | A conceptual architecture and model for smart manufacturing relying on service-based DTs | X | ✓ | X | X | ✓ | X |
| [48] (2019) | Advanced physics-based modeling approach for predictive maintenance using DTs | X | ✓ | X | ✓ | ✓ | X |
| [46] (2016) | A model-based machine predictive maintenance based on DTs and a simulation platform | X | ✓ | X | ✓ | ✓ | X |

**Table 1.** *Cont.*

| Ref | Highlighted | Blockchain | DT | Collaboration | Data Analysis | I4.0 | Transp |
|---|---|---|---|---|---|---|---|
| [47] (2018) | A modular-based corrective maintenance methodology using DTs to automate decision making in complex systems | X | ✓ | X | ✓ | ✓ | X |
| [34] (2018) | Discussion of the DT and big data in smart manufacturing in terms of applications, production, manufacturing, maintenance prediction | X | ✓ | X | ✓ | ✓ | X |
| [37] (2019) | A tool and technologies for DT in smart manufacturing | X | ✓ | X | X | ✓ | X |
| [36] (2020) | A DT-driven sustainable technique smart manufacturing | X | ✓ | X | X | ✓ | X |
| [35] (2020) | Focusing on DTs with Industry 4.0 and data analytics | X | ✓ | X | ✓ | ✓ | X |
| [52] (2018) | Investigation wind farm and power consumption for smart manufacturing using IoT and DTs to perform wind turbines maintenance. | X | ✓ | X | ✓ | ✓ | X |
| [23] (2020) | Collaboration of drone and IoT to enhance smartness of smart cities applications | X | X | ✓ | X | X | X |
| [18] (2021) | A conceptual framework for DTs collaboration to provide an auto-detection of erratic operational data by utilizing the intelligence of operational data in the manufacturing systems. | X | ✓ | ✓ | ✓ | ✓ | X |
| Our work | A conceptual framework for blockchain based DTs collaboration to improve DTs collaboration in transportation systems and focuses on real-time operational data analytics. | ✓ | ✓ | ✓ | ✓ | ✓ | ✓ |

## 3. Proposed Conceptual Framework of Data-Driven Blockchain-Based Collaborative Digital Twins

The conceptual framework of data-driven blockchain-based collaborative DTs is proposed to empower more intelligent and collaborative solutions for DTs based on DLT and distributed consensus decision-making. The proposed framework is considered one level higher than the adoption of blockchain with DT in production systems that could integrate blockchain and operational data analysis. Moreover, the proposed framework could be developed and implemented on top of the DT platform, which exploits blockchain capabilities to guarantee transparency, decentralized data storage, data sharing, peer-to-peer communication, secure and trusted traceability, and scalability.

### 3.1. High-Level Requirements for Digital Twins Collaboration

In this subsection, we have identified a set of requirements regarding DTs collaboration in distributed manufacturing. Table 2 summarizes nine criteria to fulfill the requirements for digital twins collaboration. For any manufacturing process, the data is collected from the data sources, which are DTs of physical things such as devices, machines, people . . . , etc., across the entire network within the manufacturing units. This data is frequently updated in real-time, beneficial to the organization's decision-making process (R1 & R2). Interestingly, the collected data from the collaborative DTs are data-driven learning systems. In particular, DTs provide the data analysis engine by continuously updating data fed into the learning models to enable advanced predictions of the potential risks (R3).

Basically, simulation models help to understand what may happen when changes occur on the physical assets. However, DTs help understands what is currently happening on the physical asset and what could happen in the future (R4). For the collaborative environment, the virtual visibility of the potential risks in the future within the physical assets can help to refine the product design, real-time troubleshooting, and implement new ideas. Furthermore, DTs networks are used to enhance the connectivity of the network participants (e.g., devices, machines, people, departments, organizations). The interaction between the participants through the DTs network is used for reliable data exchange to allow internal and external data sharing (R5). However, the connected DTs network needs to be authenticated to maintain trust among network peers (R6). So, authentication provides a trust level that can keep secure collaboration and interactions among the DTs network. DT is a virtual representation of a real thing and transparently visualizes the physical things and their behavior within the collaborative environment. In particular, the transparent visibility of things through the DT model allows accurate traceability across the DT network (R7).

For any centralized network, all nodes are connected under a single authority. However, the decentralized network has not a single authority server that controls the nodes, where all nodes have individual entities. Substantially, centralizing suffers from single failure data, while decentralization suffers from lacking global data, so decision-making is required for DTs collaboration (R8 & R9). Substantially, the consensus-based distributed decision-making process provides insightful and delivers efficient and reliable collaborative solutions.

**Table 2.** Requirement of digital twins collaboration.

| Req. No | Requirement | Reason |
|---|---|---|
| R1 | Data collection | supporting data-driven decision making |
| R2 | Data update frequency | providing realtime update of the physical twin |
| R3 | Data analysis | enabling advanced predictions of the potential risks |
| R4 | Simulation capabilities | enabling virtual visibility of the products |
| R5 | Data exchange | allowing internal and external data sharing |
| R6 | Authentication | maintaining trust among network peers |
| R7 | Transparency | allowing traceability across the entire network |
| R8 | Distributed decision making capabilities | providing insightful consensus-based decision making process |
| R9 | Decentralization | delivering efficient and reliable solutions |

### 3.2. Components of the Proposed Framework

The developing blockchain can overcome the safety and security that have prevented DT initiatives. Because of blockchain characteristics like immutability and decentralization, DTs initiatives can evolve more effectively and quickly in their environments [53]. A decentralized blockchain network can help trust DTs with a data track and digital identity. Furthermore, decentralization, secure and safe data transportation is more related to authenticate DTs industry environments cases. The authors of [10] introduced blockchain for DTs processing which guarantees safe, secure, and reliable transactions without data accessibility, traceability, and immutability. Smart contracts are used to track and manage the transactions in the developed DTs. Blockchain technology is used to exchange and store DTs data to exchange information between DTs in a decentralized fashion.

Consequently, to implement DTs collaboration in distributed manufacturing, the aforementioned high-level requirements could be fulfilled using a combination of DTs, blockchain, and artificial intelligence. Based on these requirements, two main components are required to equip the conceptual framework of blockchain-based DTs collaboration. Figure 4 depicts the components including (1) data-driven ledger-based collaborative DTs for predictive analytics and, (2) consensus-based decision making. These two parts will be elaborated flowingly. Furthermore, A more detailed architecture with an in-depth discussion of every component is beyond the scope of this paper.

### 3.2.1. Data-Driven Ledger-Based Collaborative DTs for Predictive Analytics

This component is used to develop a methodology for creating and updating data-driven ledger-based collaborative DTs. It demonstrates the predictive analytics approach by developing offline and online predictive models using the data ledger-based historical DTs data and live streaming DTs data. Two main sub-components are required to equip the data-driven ledger-based collaborative DTs for predictive analytics as follows:

### Ledger-Based DTs Model

Multiple DTs could be connected through the blockchain network using DLT to secure distributed operational data management and analytics across multiple participants. Figure 5 describes the ledger-based DT model. At the technical level, the ledger-based DT model needs to define the five components [10,41]: (1) registered DT owner, (2) DT status, (3) timestamp, (4) transaction, and (5) ledger database. The information that maintains a physical object's specifications is stored within the ledger, such as the DT owner. The communication mechanism that transfers bi-directional data between a DT and its physical counterpart will generate data that is considered DT status within timestamp, which is used to create a transaction. DLT is used to store the transactions, DTs data, and actions.

Moreover, DLT guarantees the transfer of values between DTs for collaborative DTs-based applications. In doing so, the ledger-based running database synchronizes the updated DTs' status within the manufacturing systems in real-time, which leads to an increase in real-time prediction accuracy, such as the potential risks to improve the quality of the decision making. The requirements of the DTs in terms of data schema and collaborations, including information and analytics sharing, will be identified based on the collaborative DTs-based applications [54]. Accordingly, the participants represented by DTs (i.e., if they are the same type or different types) should be defined. The type of collaborations in the participants' communications activities should also be determined based on the collaboration scenario from the beginning.

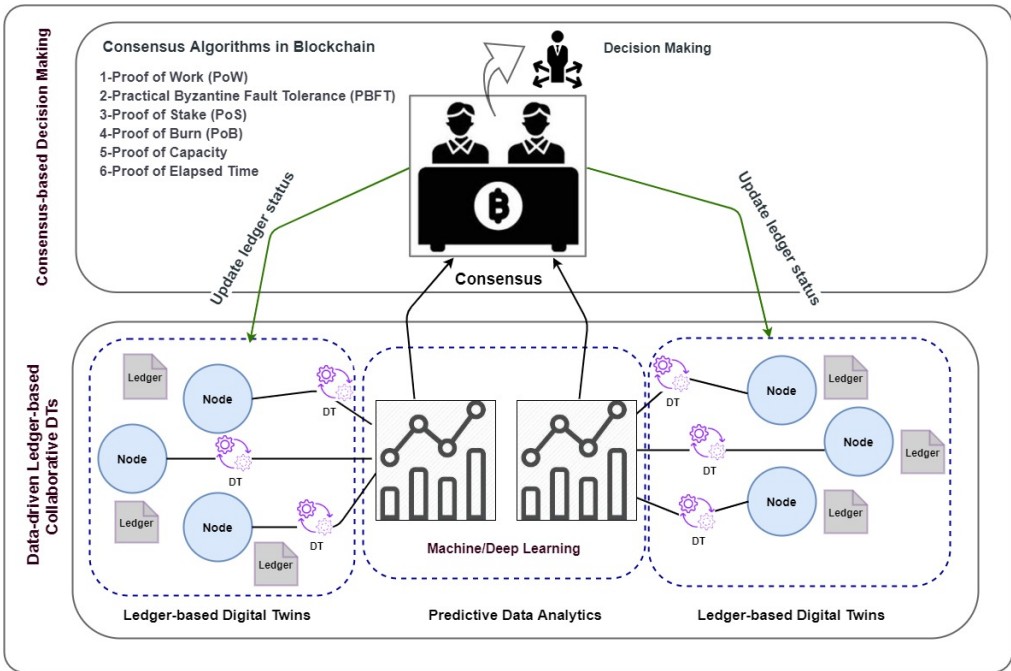

**Figure 4.** The architecture of the conceptual framework of data-driven blockchain-based for DTs collaboration.

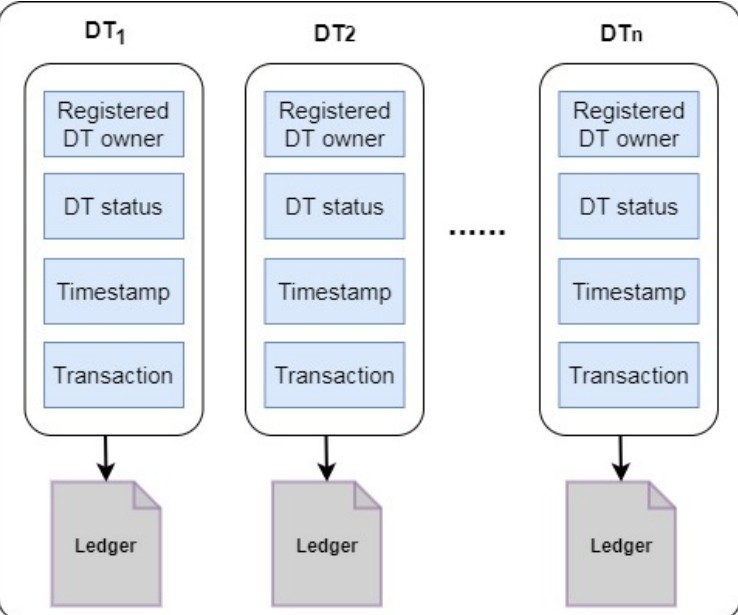

**Figure 5.** Ledger-based DTs.

Data-Driven DTs Based Predictive Analytics

For data-driven DTs collaborations, the DTs collaboration concept within the proposed framework will help to understand the DT status, interact with other DTs at the edge level, learn from other DTs, and share common semantic knowledge within industrial systems [18]. Figure 6 shows the workflow description of building a predictive model for data-driven ledger-based collaborative DTs. The workflow consists mainly of two phases: building an offline predictive model and deploying an online predictive model [55,56]. For the offline predictive model, we will use the distributed predictive model (i.e., classifier) by applying the distributed machine learning frameworks like Apache Spark MLlib. Apache Spark MLlib is a scalable library that implements many machine learning algorithms (i.e., regular machine learning and deep learning) [57]. An offline predictive model will be developed and trained using ledger-based historical DT operational data. For the online predictive model, the developed predictive model (i.e., classifier) could be evaluated in the smart contracts. In particular, the developed model will be used to predict the potential risks online using DT-driven real-time operational data. It will be run until it reaches a consensus. The smart contract will make decisions based on these output predictions and then store the decisions in the blockchain ledger. The smart contract executed the developed model and applied it using the DT-driven live streaming data stores in the ledgers. One example of the predicted potential risk with production systems is detecting the early faults indicated by degraded performance or damaged physical counterpart (e.g., node, device). Consequently, the proposed framework can help the decision-makers dynamically replan a set of safety precautions and take the proper action to decrease downtime within the production systems.

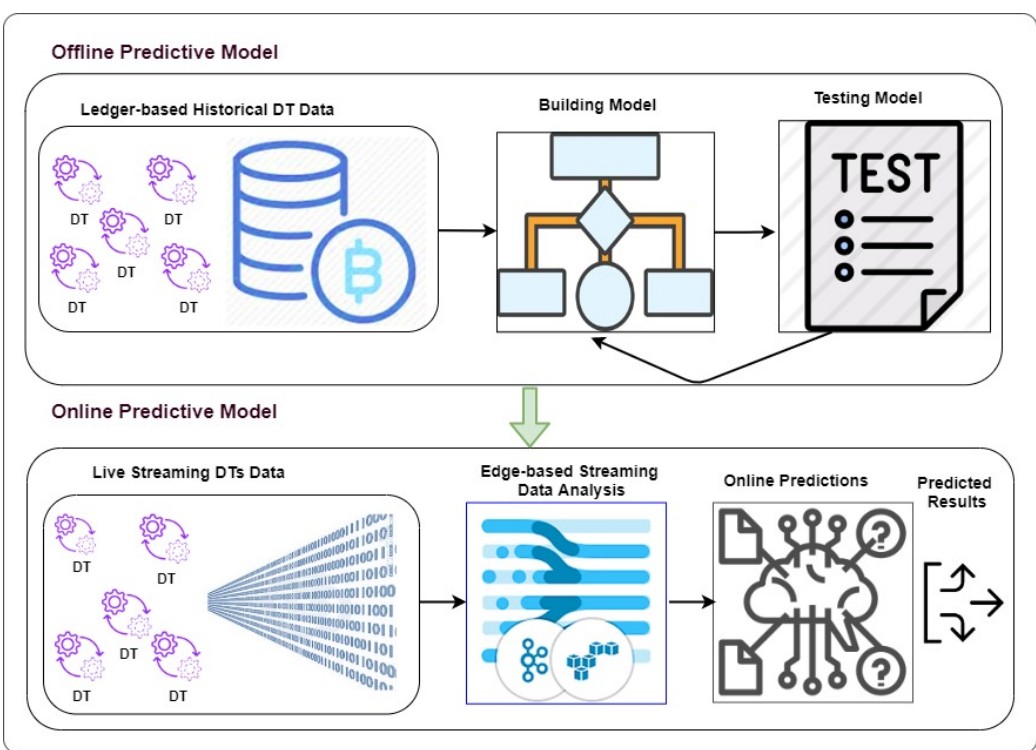

**Figure 6.** The workflow of building a predictive model for data-driven ledger-based collaborative DTs.

3.2.2. Consensus-Based Decision Making

This component is used to develop a distributed consensus algorithm to improve the DTs collaboration. The proposed conceptual framework aims to develop a collaborative DTs system to provide distributed decision-making to avoid potentially threatening the production system. Therefore, a distributed decision-making algorithm will be developed

based on the essence of the consensus mechanism and the dynamic prediction, which uses real-time DT-driven operational data [13,14]. The developed distributed consensus algorithm will provide evidence that most nodes agree on the potential risks to notify the decision-makers within the distributed manufacturing systems. Some examples of the use of the consensus algorithms include Proof of Work (PoW), Practical Byzantine Fault Tolerance (PBFT), Proof of Stake (PoS), Proof of Burn (PoB), Proof of Capacity, and Proof of Elapsed Time.

To do so, the Directed Acyclic Graph (DAG) structured DLT solution will be considered as part of its consensus mechanism that would make it possible to have a fully decentralized manufacturing environment [45,58]. Furthermore, DLT can ensure that all the participants (i.e., deployed DTs) share identical knowledge to allow the possibility of prediction that needs to reach a global agreement regarding the object of interest (e.g., fault diagnosis) [41]. This agreement will be reached using the consensus mechanism, which utilizes the local interaction of deployed DTs within a manufacturing node, and then globally using the ledger-based running database [59].

## 4. Smart Transportation Use Case

This section introduces an overview of the smart transportation industry, followed by the three selective use cases, including smart logistics, railway predictive maintenance, and their mapping to our proposed framework.

### 4.1. Overview of Smart Transportation Industry

The concept of Smart Transportation Systems (STS) includes a wide variety of advanced technologies, such as communication, sensing, and power, which have been utilized to process massive volumes of information to overcome the issues in the urban region [60]. We present an overview of the literature by considering the DTs' role and data analytics in transportation systems.

The authors in [61] have addressed the data fusion generated from the connected vehicles in STS. The authors have introduced a multi-variate data fusion (MVDF) technique. The proposed MVDF aims to handling asynchronous and discrete data from the environment and streamlining them into continuous and delay-less inputs for the applications. They have used regression learning to identify the errors, and they have used network simulator experiments for the metrics error, data utilization ratio, and computation time. In the context of the Internet of Vehicles and security, the authors in [62] have proposed two models to enhancing the security and performance of nodes on the Internet of Vehicles. The first model has been proposed to detect the data fascination attack using the hashing technique, while the second model has been proposed to reduce the travel time in case of traffic congestion. Furthermore, the authors in [63] have introduced the hybridized cryptographic-integrated steganography (HCIS) algorithm. The HCIS algorithm is used with auxiliary data inputs for secured data sharing in IoT assisted cloud environment for the urban transportation system.

In the context of logistics, the authors in [64] have proposed an IoT-assisted intelligent logistics transportation management framework to design an optimized logistics plan, improve customer service and reduce transportation costs. The authors focused on identifying the optimal routes for the directed autonomous vehicle, considering different vehicles.' necessities, renewable generations, logistic requests, and the essential transportation systems. In [65], the authors have explored the potential of using DT technology in synchromodal transport. They have introduced a proof-of-concept for long-distance DTs solution. The DT-based solution aims to connect real-time data generated from the physical system to a virtual GIS environment and then utilize this data in real-time synchromodal deliveries.

For the predictive maintenance in the transportation context, the authors in [44] have reviewed the use of DTs technologies to apply maintenance strategies to provide a deeper insight into the synergies between both DTs and maintenance in the industrial sectors. Furthermore, the authors in [1] proposed a set of requirements to enable predictive

maintenance with big data for Industry 4.0 applications. The authors have studied the railway industry concerning big data streaming processing platforms, distributed message queue management systems, big data storage platforms, and streaming SQL engines. For the railway, the authors in [66] have also explored the potential use of blockchain technology in the railway industry by providing a simple analysis of the possible adoption rate. The authors have introduced a prototype for speech recognition and a mobility-based data collection solution to enhance the technology adoption rate.

In the context of adoption DTs in crane operations, the authors in [67,68] have proposed a maintenance model called Integrated Maintenance Decision Making Model (IDMM) for cranes operating in control terminals. The IDMM model targets to improve crane operations using DTs technologies by introducing crane maintenance based on Monte Carlo Markov Chain and Particle Swarm Optimization. Furthermore, Autiosalo et al.,[69] have introduced a multi-component DT for an industrial overhead crane. The authors have proposed a prototype called Ilmatar based on DTs technologies. The Ilmatar prototype aims to provide a maintenance service for cranes' daily tasks.

On the other hand, the authors in [70] have introduced the Robotic Process Automation (RPS) solutions to deliver service for the patients. They have proposed a simulation-based framework using RPA solution, development, 3-D building information collection, supply chain simulation, and optimization. The authors have also studied the Greenfield hospital in Singapore and then used DTs technology to visualize the operational logistics supply chain. The authors in [71] have explored the potential of using DTs technologies to manage the COVID-19 pandemic by supporting flexible decision making. The authors have discussed various challenges, including modeling and data-driven analysis for pandemic management and modeling and predictive analysis. Finally, the authors have introduced a framework using DTs and AI tools to improve the control of the COVID-19 pandemic.

Table 3 describes a comparison of existing work and the present work concerning the STS use cases, including blockchain, DTs, collaboration, DT, data analysis, IoT, and the three selected transportation use cases; logistics and railway.

**Table 3.** Comparison of existing work and the present work for smart transportation use cases.

| Ref | Highlighted | Blockchain | DT | Collaboration | Data Analysis | IoT | Selected Use Cases Logistics | Railway |
|---|---|---|---|---|---|---|---|---|
| [61] (2021) | MVDF technique for handling asynchronous generated from the connected vehicles in STS | X | X | X | ✓ | ✓ | X | X |
| [62] (2021) | Enhancing the security and performance of nodes on the Internet of Vehicles. | X | X | X | X | ✓ | X | X |
| [63] (2020) | Hybridized cryptographic integrated steganography algorithm for secured data sharing in IoT in cloud environment | X | X | X | X | ✓ | X | X |
| [64] (2021) | IoT-assisted intelligent logistics transportation management framework to optimize logistics | X | X | X | X | ✓ | ✓ | X |
| [65] (2020) | Exploring potential of using DT in synchromodal transport | X | ✓ | X | X | ✓ | X | X |
| [44] (2020) | Using DTs to apply maintenance strategies in the industrial sectors. | X | ✓ | X | X | ✓ | X | X |
| [1] (2020) | Identify a set of requirements to enable predictive maintenance for Industry 4.0 including railway. | X | X | X | X | ✓ | X | ✓ |
| [66] (2018) | Exploring and analysis of adoption of blockchain in the railway | ✓ | X | X | X | X | X | ✓ |
| [67] (2019) [68] (2020) | Proposing a DT-based maintenance model for cranes operating in control terminal. | X | ✓ | X | X | ✓ | ✓ | X |

**Table 3.** *Cont.*

| Ref | Highlighted | Blockchain | DT | Collaboration | Data Analysis | IoT | Selected Use Cases Logistics | Railway |
|---|---|---|---|---|---|---|---|---|
| [69] (2021) | Proposing multi-component based on DTS called Ilmatar for overhead cranes. | X | ✓ | X | X | ✓ | ✓ | X |
| [70] ( 2020) | Proposing a robotic process automation solution to deliver service for the patients | X | ✓ | X | ✓ | X | ✓ | X |
| [71] (2020 ) | Study the potential of using DTs to manage the COVID-19 | X | ✓ | X | ✓ | X | X | X |
| [29] (2020) | A blockchain framework for heterogeneous multi-robot collaboration to combat COVID-19 | ✓ | X | ✓ | X | ✓ | X | X |
| [22] (2019) | The collaboration between drone and IoT devices for improving Industry 4.0 applications such as smart city, smart healthcare | ✓ | X | ✓ | X | ✓ | X | X |

*4.2. Selective Use Cases for Smart Transportation Industry*

Today, many transportation systems connect their information systems using new technologies, including IoT, big data, DTs, and AI. DTs are used to visualize the transportation infrastructure to support collaborations for accelerating the transportation process. In particular, DTs represent the physical assets in the transportation system to understand the assets' status and model their performances. They are continuously updated in real-time from multiple transportation systems, including sensors, vehicles, CCTV, people, road networks, etc. These DTs can be collaborated by sharing their operational data to provide insightful information about the assets throughout the lifecycle within the transportation system.

Consequently, DT collaboration is considered the backbone of the transportation system. It provides up-to-date information to implement the approaches based on predictive analytics for making decisions. In particular, decisions will be taken based on the predicted potential risks within the transportation system to avoid delays and optimize transportation asset performance.

This section introduces two use cases of smart transportation that use blockchain-based collaborative DTs: (1) smart logistics and (2) railway predictive maintenance. The proposed framework, data-driven blockchain-based collaborative DTs, could be applied in actual use cases for smart transportation. A set of concepts can be discussed using the proposed framework for designing smart transportation use cases, which are the need for DTs collaborations, DLT, predictive data analysis, and distributed consensus decision making. To guide this research work, we stated these five questions derived from the objectives of the proposed framework.

1. Why do we have to use collaborative DTs in this use case?
2. What are the data schema and requirements which DTs will represent?
3. How could a DLT be used for data sharing to support collaborative DTs?
4. How can the DTs-based operational data intelligence help gain insight to enhance the prediction about the potential risks?
5. How could a distributed consensus algorithm be used to ensure a consensus of the decision-making based on the predicted potential risks?

4.2.1. Smart Logistics

In this subsection, an overview of smart logistics is presented together with a detailed mapping of our proposed framework.

Overview

In the supply chain context, the transportation process refers to the movement of products from one location to another to deliver them. Smart logistics makes supply chain transportation more effective and efficient at each step. This means that logistics are becoming increasingly challenged, and the transport of large items is becoming a huge issue. The efficient logistics system can purchase, transportation, and store the raw materials until they are delivered, making more profits for the business and ensuring reasonable customer satisfaction about timely delivery. The connected devices in the transportation system are used to visualize the logistics process and track the movement of the products. For example, the sensors within the containers are used to track each stage of shipment, weather conditions, temperature, and humidity to give companies real-time visibility of the product movement through the logistics life cycle.

One of the state-of-the-art research works of logistics collaboration, Jabeur et al., has addressed the problem of collaboration within logistics [72]. The authors have proposed a multi-agent-based solution for collaboration between logistics objects. Three types of logistics applications could be considered: shipment alert, dynamic routing, and predictive maintenance. Dynamic routing in logistics operation is the key to a successful logistics

company. Therefore, optimizing the dynamic routing is essential in logistics for the most efficient routes allocated to delivery fleets. Due to the COVID-19 pandemic, logistics companies face challenges providing high logistics services for customers who prefer a safer and faster delivery method rather than venturing out for the same.

Consequently, the dynamic route plays a vital role in ensuring online deliveries to the quarantine areas. This motivates us to discuss the dynamic routing scenario within the logistics use case concerning our proposed framework. Based on this, Figure 7 depicts the high level of mapping of our proposed framework to logistics systems. Further details are elaborated following:

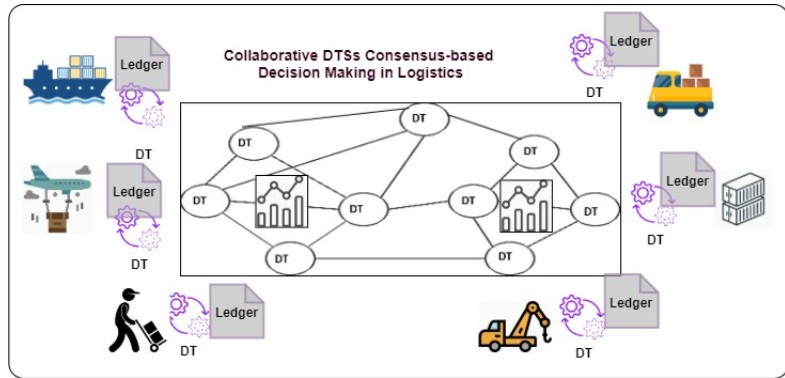

**Figure 7.** Blockchain-based DTs collaboration for logistics system.

DTs Collaboration in Logistics

Recently, Logistics 4.0 technologies have emerged as one of the dimensions of Industry 4.0, including smart robotics, self-driving vehicles, and automated systems for managing the movement of products among warehouses and factories. Logistics 4.0 solutions aim to create interoperable and connected logistics chains to become more innovative. The logistics process generates big data generated by tracking the movement of goods. According to the Industry 4.0 principles, Logistics 4.0 can be described as collaborative cyber-physical systems. Therefore, collaborative digital twins are used to represent the logistics data. The DTs-based logistics data is used for potential logistics optimization by monitoring physical assets and other equipment to eliminate downtime within the logistics system. The physical asset could be a fleet, truck, ship, container, robots, warehouse, and people with a set of sensors that can collect real-time data and operational status about the logistics supply chain. The digital logistics supply chain twin is used for end-to-end product tracking and identifying issues by visualizing goods' digital movements over time. Furthermore, the decentralized digital logistics supply chain twin comprises the DTs representing the geographical logistics warehouses, logistics centers, and participants.

In particular, the data generated from the elements within the logistics supply chain could be represented in DTs. The collaborative DTs are adopted to visualize the logistics supply chain, which could track products and provide end-to-end service from unloading at the quayside to shipping goods to their destinations. The proposed data-driven blockchain-based collaborative DTs framework will provide smart logistics service for a faster flow of goods, real-time analysis of comprehensive supply chain data, better synchronization of dynamic routing logistics processes, unbroken shipment tracking to improve distribution planning and delivery reliability.

Data Schema and Requirements for DTs

The collaborative DTs model consists of three components; (1) digital model, (2) data analytics, and (3) knowledgebase. The logistics data within the digital supply chain is generated from sensors attached to containers, fleets, warehouses, and robots to capture real-time data about logistic items and report on-time data about environmental changes. For warehouse management, the sensors monitor the weather, e.g., temperature and

humidity in the warehouse for storing safety items. For containers, sensors are used to monitor the environmental conditions during deliveries. For fleet management, sensors attached to ships, trucks, buses, airplanes are used to collect the operating parameters of the fleet. Finally, for logistics workers, sensor data are used to track staff's physical safety.

DLT and Logistics Operational Data Sharing

The complexity of the logistics system comes from the distribution of goods, from raw materials to finished products. The logistic supply chain is divided into hundreds of geographically distributed stages globally, among multiple warehouses and logistic centers. These distributed logistic processes have massive data, making monitoring and analyzing targeted actions in response difficult. Another relevant aspect of complexity is represented by the security, integrity, sharing, and interoperability of logistic data sources. Consequently, DLT can implement collaborative DTs that allow data sharing among multiple DTs in a decentralized supply logistic chain. Therefore, the ledger-based data sharing for DTs collaboration will be considered for DT data, sharing data, and collaboration-by-communication within logistic participants.

Data-Driven DTs Based Predictive Analytics

With dynamic routing, logistics systems have a flexible and powerful scheduling capability to deliver the products to the customers on time to keep their reputation and quality of services aligned with the customers' stratification. The logistics companies can use real-time information such as weather or construction delays to change carrier routes on the fly. They can plan for future shipping routes based on collected historical logistics operational data. They can also increase their profits by taking place the dynamic routing to continue delivering a flawless shipping experience.

To assess the potential risks within the decentralized logistics manufacturing (e.g., whether it is harmful to the products such as medicines and frozen food), a prediction is needed to estimate how long a refrigerated truck will require to arrive at one or more processing plants. Therefore, dynamic routing is essential to direct the fleets based on past experiences and real-time tracking of the on-road performance of the fleet. When any problem occurs due to weather or roadblocks, the dynamic rerouting feature helps decision-makers suggest alternate and efficient routes for delivery. Consequently, many variables are operating within trucks that are needed to be monitored (i.e., temperature and humidity inside the containers, driver's road time, and the route conditions). The data generated from the elements regarding the refrigerated truck could be represented into DTs at hierarchical levels: (1) With local DT at each container, (2) intermediate more powerful DTs on the refrigerated truck at the network edge, and (3) much more powerful when DTs represent the logistics units in the cloud. The refrigeration unit's collaborative DTs system can predict the product state and queue length using the DT-driven operational data. Based on these predictions, the decision can be made in a consensus-based manner to direct the truck to the best plant to avoid potential risks.

Consensus-Based Decision Making

A consensus is a decision-making process in which members of a group of logistics centers agree to develop and support decisions to speed up the logistics supply chain considering mutual logistics traceability. Using collaborative DTs provides a better understanding of potential risks for logistics supply chains and facilitates consensus-building among participants involving the decision-makers. In doing so, many nodes involved in logistics information that is represented in DTs are divided into multiple consensus sets. The consensus mechanism which could be used is the PBFT algorithm. According to the PBFT algorithm, some amount of fault (damaged objects, hijacking, and theft, climate change, fleet failure, CCTVs failure, staff's physical safety, transportation conditions ) can be tolerated without affecting the integrity of the network. The PBFT algorithm is used

in Hyperledger in the transaction approval process to avoid malicious decisions among participants with logistics supply chains [73].

### 4.2.2. Railway Predictive Maintenance

In this subsection, an overview of predictive railway maintenance is presented together with a detailed mapping of our proposed framework.

#### Overview

Railway 4.0 is one of Industry 4.0 dimensions using new digital technologies including big data, IoT, DT, AI, and cloud computing [1,74,75]. The railway companies compete to provide high and attractive service to the passengers by utilizing automation and emerging technologies. One of the significant challenges that the rail industry faces is avoiding delays to meet passengers' satisfaction and maximize their profits. To do that, the rail companies start to deploy predictive maintenance applications to early diagnoses the fault and perform maintenance actions.

Nowadays, railway companies use DTs to improve railway performance by utilizing railway DT-based operational data analysis intelligence. In particular, they use the DTs collaboration to gain improved information visibility and better understand the past, present, and future predictions. With the DT-based prediction of the potential risks, the decision-makers in rail companies can support the transformation of rail track maintenance and deliver safe, reliable, and resilient service.

As the result of digitalization in the railway sector, blockchain has been adopted to provide security, scalability, traceability, transparency, and decentralization [66]. However, the adoption rate of the blockchain can be seen as slow due to the lack of technology stability, maturity, and developers' skills. Despite that, the combination of blockchain and DTs technologies in the railway sector has a beneficial role in providing high quality and safe service by (1) representing the complete railway supply chain, (2) automating of internal accounting process and passenger traveling, (3) Conducting contracts between machines and objects, and (4) monitoring the railway assets including station, train, track, switches, point machine, and sensors, and (5) managing signaling, passenger information systems, physical flows, ticketing, and goods delivery. One of the biggest challenges for the rail companies is making their stations service with a minimum maintenance cost by avoiding unnecessary expenses for the maintenance company. As the late railway maintenance can result in failure and additional costly repair, using simulation by utilizing DTs capabilities, predictive data analysis, and blockchain technologies can capture the early fault and notify the decision-makers to take the proper actions. Based on this, Figure 8 depicts the high level of mapping of our the proposed framework for fault diagnosis in railway systems. Further details are elaborated following. Furthermore, Table 4 summarizes a comparison of some current research work in predictive maintenance.

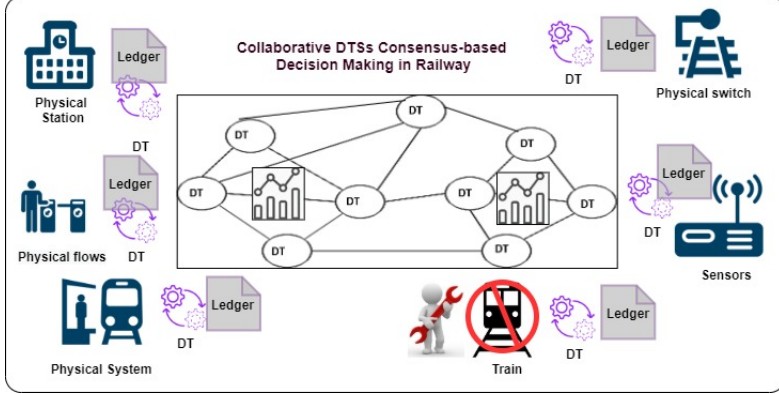

**Figure 8.** Blockchain-based DTs collaboration for railway.

**Table 4.** Comparison of some existing research work in predicative maintenance.

| Ref | Highlighted | Technologies | Advantages | Limitations |
|---|---|---|---|---|
| [1] (2020) | Identify a set of requirements to enable predictive maintenance for Industry 4.0 including railway | Big data streaming technologies including -distributed queuing management, -big data stream processing, -big data storage, -streaming SQL engines | Provide a breadth-first mapping of predictive maintenance use-case requirements to the capabilities of big data streaming technologies focusing on open-source tools | Using the capabilities of blockchain and DTs Collaboration |
| [74] (2016) | Discussing the possibility of applying predictive maintenance in the railway transportation industry | -RFID technologies -Business intelligence | Explore the most important positive effects of applying predictive maintenance that affect the organization, the economy, and the people in the whole system | Using the capabilities of blockchain and DTs Collaboration |
| [75] (2018) | Proposing a cloud-based system for cost-effective and reliable real-time data collection, processing, and analysis from the shop floor with a multi-criteria decision-making algorithm and a condition-based maintenance strategy | -Cloud computing -Wireless sensor network -Data acquisition technologies | -Increasing awareness on both machine and shop-floor level condition -Effective and accurate maintenance of machine tools -Accurate decisions though condition-based maintenance and adaptive scheduling. -Increasing interoperability, automation and communication | Using the capabilities of blockchain and DTs Collaboration |

**Table 4.** *Cont.*

| Ref | Highlighted | Technologies | Advantages | Limitations |
|---|---|---|---|---|
| [76] (2018) | Reviews the overall framework to develop a DT coupled with the industrial IoT technology to advance aerospace platforms autonomy | -Industrial IoT<br>-DTs | Discuss the role of data fusion in predictive maintenance using DT | Using the capabilities of blockchain and DTs Collaboration |
| [77] (2020) | Investigating of creating an automation cell for the fan-blade reconditioning component of maintenance, repair, and overhaul services to ensure that fleets of aircraft are in airworthy conditions | -Vision sensor<br>-DTs<br>-Robotic technologies | Track and remove the coating material of a fan blade in a closed-loop approach | Using the capabilities of blockchain and DTs Collaboration |

DTs Collaboration in Railway

The rail sector is a complex network of assets and systems that come together to enable people and goods to travel safely, in a timely way at various speeds and distances. The rail assets are including station, process, rail system, and individual assets such as train, switch, sensors, etc. These rail assets could be represented in interoperable and collaborative DTs to show high visibility of the rail supply chain. The DTs representing the rail supply chain of the transportation industry can collaborate to diagnose the railway's fault, whether caused by hardware failures (e.g., rail, train, switch) or weather conditions that affect the rail, for example temperature and humidity. The DTs collaboration can understand each DT status, interact with other DTs, learn from other DTs, and share common semantic knowledge across geographical railways.

Data Schema and Requirements for DTs

In the rail industry, the physical assets (e.g., trains, track, switches, point machines) generate a vast amount of machine data from sensors such as temperature, light, vibration, and GPS. The rail data can be used to identify potential railway failures. To do so, the physical rail assets are being monitored by collecting their sensory data. As DTs represent the railway industry, the DT model is defined based on the data required for data-driven analytics for fault diagnosis, holding data fields that could be fitted in the predictive data models. Consequently, the collaborative DT model is defined based on the basic DT model, and it is consists of three components, including digital model, data analytics, and knowledgebase [78]. These components are integrated to investigate the DTs collaboration, locating the fault within the rail industry.

The Digital model of the operational data contains semi-structured content, e.g., JSON and XML. According to the domain of the rail industry, the data is generated from rail assets such as station, process, train, and weather sensors. For example, the rail data describes the physical system of the rail station, and the weather sensors are used to monitor climate change. Their data represent the weather variables, including temperature and humidity. Furthermore, the model is used to semantically model the data, reflecting the DT features and their relations using object-oriented concepts. Some semantic work could describe the relationships between the models by using model-to-model, e.g., OOP, RDF, and OWL in the case of the complex DT systems with heterogeneous DTs types.

DLT and Railway Operational Data Sharing

Distributed ledgers provide a novel technology for multi-party data sharing that emphasizes data sharing and integrity. To implement a collaborative DTs system that allows data sharing among multiple DTs in a decentralized railway supply chain, we consider DLT. The proposed framework for DLT-based architecture for DT data sharing to support DTs collaborations [41] could be extended to adopt DTL for collaborative DTs. Regarding data sharing, both the communication between lifecycle parties and the bidirectional communication between the DT and its real-world railway asset counterpart need to be considered. Therefore, the proposed ledger-based data sharing for DTs collaboration should consider: (1) DT data, (2) sharing data, and (3) collaboration-by-communication. Regarding using the DLT for the rail industry, each ledger should share data for the rail station with all parties.

Data-Driven DTs Based Predictive Analytics

Predictive data analytical models are used to support decision-making by utilizing the intelligence of the DTs-based operational data. For instance, data analytics can be used within DTs' interaction and communication to describe, diagnose, predict, and prescribe the behavior of the physical rail system for fault diagnosis. The outcome of the data analytics will be used as inputs for the consensus algorithm to make the best decision for abnormal

data or warn the decision-makers in case of potential failure over the rail industry supply chain. The knowledgebase also contains the set of knowledge learned through relevant machine learning techniques from historical maintenance.

The component of data-driven ledger-based collaborative DTs for predictive analytics in our proposed framework can perform the potential failure risk using rail-based operational data including heterogeneous data sources (i.e., sensor data and historical data) [79]. To do so, the offline predictive model will be trained using ledger-based historical DT operational data. For the online predictive model, the developed predictive model (i.e., classifier) could be evaluated to predict the future faults with the railway. The outcome of the data analytics will be used as inputs for the consensus algorithm to make the best decision for abnormal data or warn the decision-makers in case of potential failure over the rail industry supply chain.

Consensus-Based Decision Making

The consensus algorithms will be used to provide the agreement of the diagnosed fault provided by collaborative DTs in the railway. This agreement will be reached locally utilizing the interaction of deployed DTs within a railway and then globally using the running database within the distributed ledger, which synchronizes the updated DTs' status within the supply chain [54]. The average consensus about the fault within the rail (e.g., fleet, train, station), which relies on a belief consensus technique, will advise the decision-makers about the fault within the rail industry supply chain.

## 5. Discussion, Validation, and Future Direction

In this section, we discuss the validity of our proposed framework concerning the requirements and future directions.

### 5.1. Validation of the Proposed Framework

This section validates the proposed framework, which aims to apply a data-driven blockchain-based for DTs collaboration. For this purpose, we also consider who the aforementioned high-level requirements were fulfilled by using the industrial technologies and informative concepts. Table 5 shows the overview of mapping the industrial technologies to the identified requirements for the proposed framework. For the data collection, IoT technologies are used to allow various data sources, such as physical things like devices, machines, people, etc. These collected data are stored into DTs, which are considered as the image of physical things. These data are also frequently updated to inform the current status of the physical things. To consider this rapid update of the physical things, the concept of timely updating can be offered by using DTs technology. DTs technology can provide an AI-based system for the data analysis requirement by continuously updating data to give timely predictions that help the decision-making process. Furthermore, the DTs technology has been adopted for its dynamic simulation capability to understand what is currently happening on the physical asset and what could happen in the future.

Besides that, the update frequency of the data needs to be exchanged among the DTs network in a secure, trust, authenticated, and transparent process. Furthermore, collaboration means sharing and exchange information among entities and share tasks to act accordingly. Blockchain technology is beneficial for DTs collaboration to (1) maintain the trust among peer to peer network [10], which DTs represent, (2) allow traceability across the entire DTs network [7], (3) provide insightful consensus-based decision-making process [80], and (4) deliver efficient solutions by utilizing the decentralization feature of blockchain technology [81]. Additionally, with a decentralized infrastructure of physical nodes represented in DTs, the blockchain particularly, DLT technologies can help relieve the risk of the point of failure. The blockchain and DLT technologies can overcome the safety and security that have prevented DT initiatives. The decentralized blockchain network can help trust DTs with a data track and digital identity. For a reliable decision-making process, the Consensus algorithms are used to improve the DTs collaboration in terms of the

agreement of the majority of nodes about the potential risks to notify the decision-makers within the distributed manufacturing systems.

**Table 5.** Validation of the proposed framework.

| Req. No | Requirement | Enabled by |
|---|---|---|
| R1 | Data collection | IoT technology |
| R2 | Data update frequency | DT technology |
| R3 | Data analysis | AI techniques |
| R4 | Simulation capabilities | DT technology |
| R5 | Data exchange | Blockchain DLT technology |
| R6 | Authentication | Blockchain technology |
| R7 | Visibility and transparency | Blockchain technology |
| R8 | Distributed decision making capabilities | Consensus algorithms |
| R9 | Decentralization | Blockchain DLT technology |

Validation of the Smart Logistics Use Case

A usual way to assess the validity of the conceptual frameworks is to define a set of criteria and then compare the capability of the proposed framework with the specified criteria. The criteria used above (see Table 5) may serve as a starting point. As the proposed conceptual framework implementation is a work in progress, the smart logistics use case scenario has been obtained to be validated [72]. Concerning that the proposed framework is implemented-dependent, Figure 9 describes the workflow of blockchain-based DTs collaboration in the logistics system. Being implementation-dependent, the participants in the blockchain network of the logistics system (i.e., represented in DTs) are detailed as follows:

- The factory is responsible for transporting the loads to the suppliers. Each factory DT checks the smart contact to meet the load requirements. Then factory DT has all the data about the loading status, certificates, suppliers' locations, and the number of batches written into the ledgers. Once the factory sends the load to the supplier, the blockchain network is updated.
- The supplier is responsible for transporting the load to the warehouse. Each supplier DT has frequently updated data about the loaded products, warehouses locations, and shipments date. The collected data about the products and shipments are written into ledgers, and then the blockchain network is updated.
- The logistics operator is responsible for updating all necessary records, including a packing list, order number, batch number, production data, etc. Each logistics operator DT has frequently updated data about the corresponding shipment data recorded by the operator. The recorded data are written into ledgers, and then the blockchain network is updated.
- The long haul carrier is used to carry the heavy shipment and transport them to the warehouses. Each carrier DT checks the smart contact to meet the rules of shipment transportation. As a result, the carrier DT frequently has the updated bill of loading, shipment details, the destination warehouses location, and diver details. The shipment data are written into ledgers, and then the blockchain network is updated about the shipment movement track.
- The warehouse is used to store the shipments. Each warehouse DT has the data about the stored products, including location, temperature, humidity, product items, etc. These updated warehouse DT data are used to check the product storing conditions

concerning the smart contract rules to avoid product damages. The warehouse DT data, including the product quantity, are also used to check the smart contract for new orders. The stored product data are written into ledgers, and then the blockchain network is updated about the stored products.

- The delivery process is responsible for delivering the orders to the customers. Each delivery DT has updated data about the warehouse location, customer address, routing instruction, packing details, driver details, bills, etc. The delivering product data are written into ledgers, and then the blockchain network is updated about the products being delivered.

- A customer is a person who orders the product and who receives it. The customer DT has the data about the customer delivery address, customer ID, etc. The smart contract will check the delivery data based on the customer data, and then the delivering information is recorded into the ledgers. Once the delivery process is successfully completed, all the blockchain network participants are updated about completing the delivery process.

- The decision-making unit is responsible for deciding in case of potential risk for the products during the loading, storing, and delivering processes.

Based on the criteria above and the participant in the blockchain network, the validity of the proposed conceptual framework are discussed as follows:

**Tracking**. Blockchain network allows efficient tracking of the changes along the logistics process. Using the combination of emerging technologies like the blockchain, AI, and DTs can improve the productivity of the logistics process with effective tracking. The proposed framework can incorporate blockchain with DTs to get accurate data about every step in the shipping process. Once the products are loaded and shipped, the logistics participants' DTs collaborate to exchange the logistics data. The logistics data is stored along with the information on the product movement [65]. Therefore, the blockchain network can provide the participants with the logistics by-product data like showing the person handling the product at that time. For example, the logistics system can track the product damage using the ledger-based logistics records in case of product damage. The smart contracts are settled, the logistics data will be stored in the public ledgers. All logistics records are stored to track the changes (i.e., what is the change, why it is done, and who made the changes). Furthermore, sharing information about logistics tracking with the customers across the blockchain network can increase the transparency of the logistics system.

**Delivery**. To assess the potential risks within the decentralized logistics manufacturing delivery process (e.g., whether it is harmful to the products such as medicines and frozen food), a prediction is needed to estimate how long a refrigerated truck will require to arrive at one or more processing plants. The proposed framework can incorporate blockchain with AI and DTs to predict the potential risks in advance and then take the appropriate action like routing redirection [64]. On the other hand, the proposed framework can improve the secure delivery process by reducing fraud and theft issues. To do this, the smart contracts check the detailed rules, such as requiring government-approved photo IDs to access the goods for pickup or delivery.

**Performance monitoring**. Based on the components of the proposed framework (see Section 3.2), the predictive data analysis component is used to monitor the performance of the logistics process by analyzing the product data which are collected from logistics participants DTs including factory DT, supplier DT, long haul carrier DT, logistics operator DT, warehouse DT, and delivery DT. These DTs are collaborating and interacting to feed the learning models to predict the potential risk of the product, such as harmful product, damage, theft, and so on. The decision is making based on the consensus to avoid the risk such as logistics process delay.

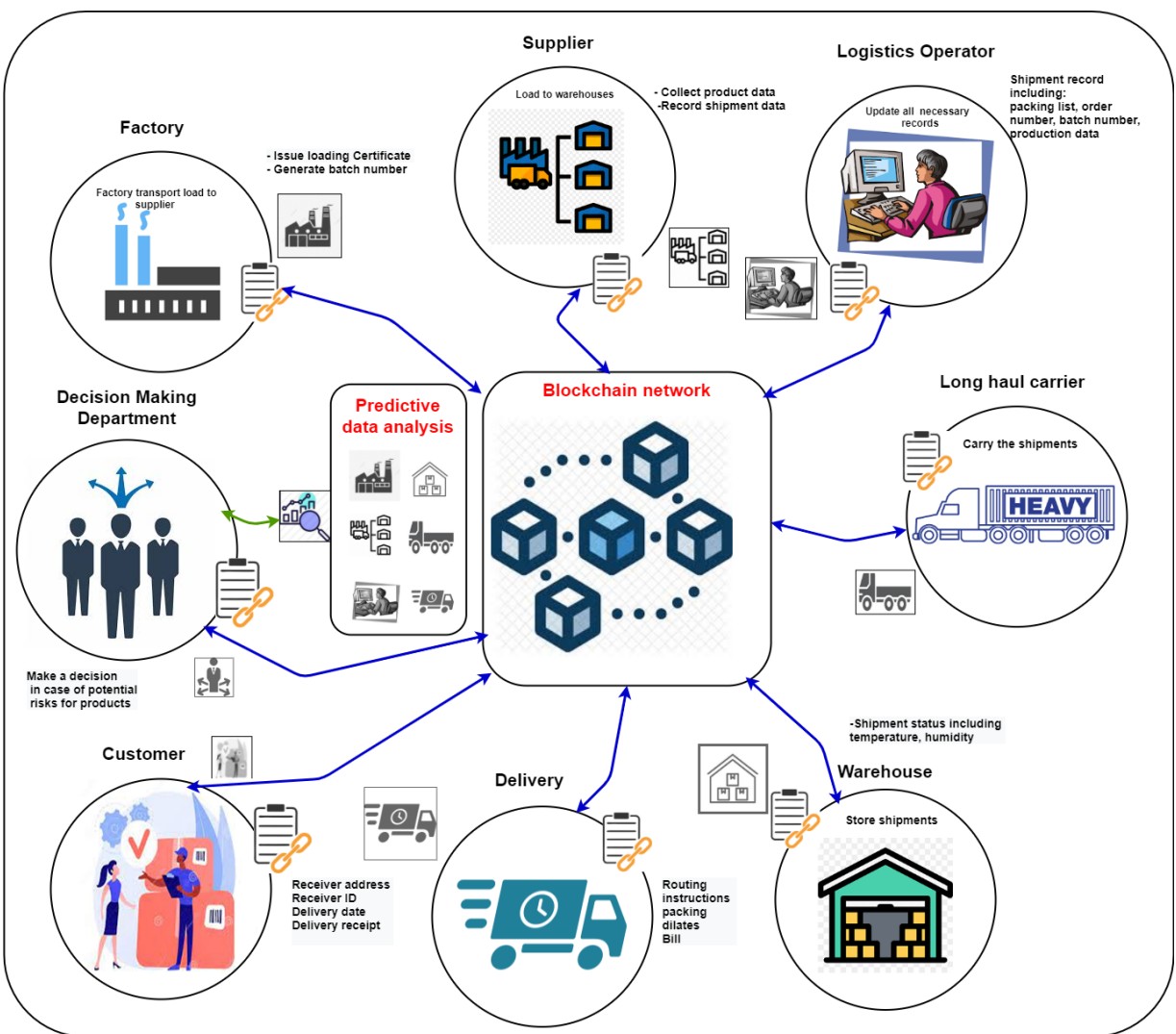

**Figure 9.** The workflow of blockchain-based DTs collaboration for logistics system.

### 5.2. Future Directions

#### 5.2.1. Security and Privacy

The security and privacy associated with DTs are challenging within smart transportation because of the massive amount of data and the risk of sensitive data created from smart transportation systems. Therefore, IoT devices should analyze DTs data locally using federated learning and then share only the model to the blockchain instead of sending the raw data. Thus, the issue of security can be solved by using blockchain technology, while privacy can be solved by using federated learning. The combination of both techniques can significantly enhance the security and privacy of DTs in transportation systems.

#### 5.2.2. Connectivity

With the growth of smart devices in smart transportation systems, connectivity is still a challenge for these smart devices to perform the goal in real-time. The massive number of smart devices in smart transportation needs for advanced communication technologies like Beyond Fifth Generation (B5G) or Sixth Generation (6G). If any smart devices get disconnected, blockchain may help devices borrow data from neighboring devices to keep the transportation system working efficiently. Running machine learning at the edge may ensure full connectivity, high accuracy and prevent missing data.

### 5.2.3. Global Logistic Networks

DTs in logistic networks play a vital role in improving logistics such as highways, railways, streets, oceans and smart cars can create their data from surrounding resources and make their map accordingly. They can then share their data with others to reduce traffic. The shared data can be traffic speed, parking, road closures, etc. and blockchain technology collaborates with them quickly in the decentralized network. While federated learning can improve processes locally and sharing only the model, this will help solve blockchain storage issues and resources. Therefore, digital twins of smart cars can collaborate in the real-time location of a specific car or people with the help of blockchain and federated learning for creating compelling, optimize, and efficient logistics.

### 5.2.4. Timing, Speed, and Response

Beyond logistics and transportation, DTs timing and speed are challenging. Timing and speed can lead to several changes in logistics and transportation. First, time improves decision-making and response of taking decisions and taking actions for the customer service demand, which needs high accuracy and fast responses. Businesses based on transportation and logistics do not want data, but they want visibility to be quick and timely.

### 5.2.5. Packaging and Containers in Transportation

In logistics, most of the transportation is in the packaging form. Therefore, the development, management, and monitoring of containers and packaging face many logistics sector challenges. Currently, due to COVID-19, increasing demand for containers and packaging can be noted due to growth in E-commerce. Furthermore, varieties of packaging also need to be taken into consideration. These results significantly reduce operation efficiency and waste due to poor utilization. A container model can be created using DTs in conjunction with computer vision, and such problems can be automatically detected. Historical information storage in the blockchain of the containers moments for starting DT can influence decision-making about container status. The decision-making should be repaired, used, and maintained as a fault in the container. DTs with blockchain technology can develop a lighter, more robust, and eco-friendly environment for packaging goods and managing containers effectively and efficiently.

### 5.2.6. Decision-Making Process

The proposed framework work could be extended by integrating two levels of the decision-making process to derive alternative system configurations. The decision-making process levels are 1) decision to avoid the potential risks and 2) dynamic planning to reconfigure the system in cases of unexpected events. For example, in the case of the railway transportation industry, the system can predict failure then diagnose and trigger maintenance by using IoT. In another instance, in the transportation logistics industry, the system can predict the potential risks of the product state within a truck. Based on these predictions, the decision can be made in a consensus-based manner to direct the truck to the best plant to avoid potential risks.

### 6. Conclusions

This conceptual framework shows how blockchain technology-empowered DTs collaboration in smart distributed manufacturing. DTs collaboration supports the interaction mechanism to understand the DT status, sharing a goal, exchange information, interacting with each other, mutual learning, and mutual adaption. Based on the literature, we present the challenges that DTs collaboration is suffering and then how blockchain technology solves these challenges. The proposed framework can improve DTs collaboration and analysis data in real-time. Furthermore, we discuss how the conceptual framework can be applied in smart transportation, i.e., smart logistics and railway predictive maintenance. Finally, we highlighted the future direction to guide interested researchers in this interesting area.

**Author Contributions:** R.S.: Conceptualization, Data curation, Formal analysis, Methodology, Writing—original draft, Writing—review & editing. S.H.A.: Conceptualization, Methodology, Formal analysis, Writing—review & editing. Brown, K.N.B.: Conceptualization, Supervision, Investigation, Project administration, D.O.: Supervision C.M.: Conceptualization, reviewing and Project administration M.G.: Conceptualization and reviewing. All authors have read and agreed to the published version of the manuscript.

**Funding:** This research has emanated from research supported by a research grant from Science Foundation Ireland (SFI) under Grant Number SFI/16/RC/3918 (CONFIRM), and Marie Skłodowska-Curie grant agreement No. 847577 co-funded by the European Regional Development Fund.

**Institutional Review Board Statement:** Not applicable.

**Informed Consent Statement:** Not applicable.

**Data Availability Statement:** No Data available online. For further query email to corresponding author (rsahal@ucc.ie, radhya.sahal@gmail.com).

**Conflicts of Interest:** The authors declare no conflict of interest.

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
