# Peer review of "Blockchain-Empowered Digital Twins Collaboration: Smart Transportation Use Case"

_machines, doi:10.3390/machines9090193_

Round 1

Reviewer 1 Report

Dear authors,

A conceptual framework based on the Block Chains (BC) and Digital Twins (DT) technology was proposed in the manuscript, which described the application philosophy of the above two technologies in smart logistics within the Intelligent Transportation System (ITS). The conceptual framework may provide a certain reference for the application of BC and DT technologies in the ITS. However, there are many defaults existing in the manuscript, and addressing these problems is the prerequisite for the manuscript to be published in any journal. Some of the vital items are as follows:

  1. In the manuscript, there is few specific model or method and the corresponding theoretical derivation as well as scientific verification. In view of the type of the manuscript is “Article”, this is unacceptable.
  2. The application scenarios of BC and DT technologies in the ITS are described with the proposed conceptual framework. But these discussions are only kept in writing stories, and the essential theory, model, and method to support the description are absent. And a presentation on the global structure of the proposed framework should be provided.
  3. There is a large number of grammars, format, and other expressions in this article, resulting in poor readability of the article. These problems must be carefully revised, and a comprehensive proofreading by a native English-speaking expert in related fields is suggested. Some of the defaults are outlined, but the problems are not limited to these.

1) Check the title and format of all Tables.

2) Check the size and readability of all Figures.

3) Check font and size of all titles.

4) Check the citation order of references.

5) Check the layout of page 13.

6) Check line 247.

7) Check line 627.

Sincerely,

The reviewer.

Reviewer 2 Report

The manuscript explains the concepts in very vague way, so it is not clear what is the original contribution. 

The concepts are not supported by any analytical or quantitative analysis, therefore it is difficult to evaluate the validity of the concepts. 

If blockchain technology really solves the issues pointed out by the author, then what is the evidence to support the claim?
It is claimed that proposal improves DTs collaboration in transportation, I would like to know how efficiently and up to what extent 

Reviewer 3 Report

The manuscript analyzes digital twins (DTs) to prevent risks in distributed manufacturing systems and reach consensus decision making. The authors propose combining blockchain and DT technologies with significant advantages for DTs from a collaborative point of view for intelligent transport in decentralized peer-to-peer networks. They use the blockchain system (with features such as interoperability, authentication, distributed decision-making, scalability, and robustness) using DTs-driven operational data and develop a distributed consensus algorithm to improve the decentralized DTs collaboration with the help of machine learning predictive models. The conceptual proposal is exciting. Here are my comments:

  1. From the point of view of its conceptual proposal framework for

data-driven ledger-based collaborative DT, how do you justify always guaranteeing decentralization?

  1. I recommend that the authors attach some references on the conceptual basis of the value of the blockchain, such as Valdivia, L. J., Del-Valle-Soto, C., Rodriguez, J., & Alcaraz, M. (2019). Decentralization: The failed promise of cryptocurrencies. IT Professional, 21 (2), 33-40.
  2. It is essential to clarify the concept of predictive maintenance and the algorithms applied to the industry. Make a comparison table, for example.
  3. Regarding Smart Transportation Industry, an exciting topic is related to leveling the playing field. Many examples illustrate how blockchain technology will benefit from limited resources by providing them with faster payment, more expeditious claims handling, and more accessible bid opportunities. They could develop this idea more fully.
  4. Blockchain is used as a technological paradigm still under exploration for different productive activities, with which intermediaries are eliminated, at the same time that it allows creating distributed trust among its users thanks to the security and transparency it offers. The authors could further develop the concept concerning some examples for international trade in maritime freight and container traceability, inventory financing using IoT and BigData, and the development of secure applications for logistics and container shipping.

Round 2

Reviewer 1 Report

Dear authors,

A conceptual framework based on the Block Chains (BC) and Digital Twins (DT) technology was proposed in the manuscript, which described the application philosophy of the above two technologies in smart logistics within the Intelligent Transportation System (ITS). Overall, the revision made by the authors did improve the manuscript in the terms of grammar, format, etc. However, the essential problem has not been solved. The manuscript is merely kept in writing stories, of which parts of the contents are common on the Internet. Considering that the type of the manuscript is “Article”, it is unacceptable that few specific model or method is provided. It is for the editor to decide this level of importance makes the manuscript suitable to publish as an “Article” in “Machines”.

Sincerely,

The reviewer.

Reviewer 2 Report

The author has provided several benefits of his approach, but does
not provide any evidence to support them, for example, the author says the concept "deliver efficient and reliable solutions by utilizing the decentralization feature of blockchain technology" the word "efficient" is not quantified, similarly the word "reliable" is not quantified.

Just saying that the approach is efficient and reliable without providing any graphs, charts, or evidence, is not enough.  

Round 3

Reviewer 1 Report

Dear authors,

A conceptual framework based on the Block Chains (BC) and Digital Twins (DT) technology was proposed in the manuscript, which described the application philosophy of the above two technologies in smart logistics within the Intelligent Transportation System (ITS). The manuscript was still merely kept in writing stories, of which most parts of the contents are available on the Internet or in other published papers. And only some basic concepts were introduced and some relevant references were reviewed in the manuscript. However, there was no specific theory, method, or model proposed or improved in the manuscript, which is unacceptable for a research article.

Sincerely,

The reviewer.

Reviewer 2 Report

The research neither provides any "Quantitative Analysis" (i.e. any charts tables, graphs to show significant improvement over past techniques) nor any "Evidence" (i.e. graphical comparison) to support the claims made by the authors.